# General Anesthesia in Psychiatric Patients Undergoing Orthopedic Surgery: A Mechanistic Narrative Review—*“When the Brain Is Unstable, Keep It Awake”*

**DOI:** 10.3390/reports8040263

**Published:** 2025-12-12

**Authors:** Ahmed Adel Mansour Kamar, Ioannis Mavroudis, Alin Stelian Ciobica, Daniela Tomița, Manuela Pădurariu

**Affiliations:** 1Medical Department, Gulf of Suez Petroleum Company (GUPCO), Cairo Office, Cairo 11511, Egypt; 2Department of Orthopedics and Traumatology, Clinical Recovery Hospital (Recuperare), 700661 Iași, Romania; 3Doctoral School of Biology, Faculty of Biology, “Alexandru Ioan Cuza” University of Iași, 700505 Iași, Romania; 4Neurosciences Department, Leeds Teaching Hospitals NHS Trust, Leeds LS1 3EX, UK; 5Laboratory of Neuropathology & Electron Microscopy, School of Medicine, Faculty of Health Sciences, Aristotle University of Thessaloniki (AUTH), 54124 Thessaloniki, Greece; 6Department of Biology, Faculty of Biology, “Alexandru Ioan Cuza” University of Iași, 700505 Iași, Romania; 7“Ioan Haulica” Institute, Apollonia University, Păcurari Street 11, 700511 Iași, Romania; 8“Socola” Institute of Psychiatry, 700282 Iași, Romania

**Keywords:** general anesthesia, regional anesthesia, psychiatric disorders, orthopedic surgery, hip fracture, postoperative delirium, redox imbalance, vulnerable brain, perioperative neurocognitive disorders

## Abstract

Orthopedic and lower limb fracture surgeries are among the most frequent emergency procedures and are commonly performed under general anesthesia (GA). Background and clinical significance: Epidemiologically, postoperative coma after GA is rare (0.005–0.08%), but delayed awakening (2–4%) and postoperative delirium or postoperative cognitive dysfunction (POCD) (15–40%) remain significant. These neurological complications increase markedly in vulnerable brain patients with psychiatric, cerebrovascular, or neurodegenerative disorders. Methods: This mechanistic narrative review synthesizes evidence from clinical and experimental studies (1990–2025) comparing the effects of general versus Regional (RA)/local (LA) or spinal anesthesia in vulnerable neuropsychiatric populations “with pre-existing brain illness” undergoing orthopedic surgery. Domains analyzed include neuropsychiatric medications effects and interactions with the GA process and with general anesthetic agents, alongside alterations in neurotransmitter modulation, cerebrovascular autoregulation, mitochondrial dysfunction, oxidative stress, redox imbalance, and neuroinflammatory activation. The review summarizes evidence on how the choice of anesthesia type influences postoperative brain outcomes in patients with known neurological conditions. Results: From previous studies, patients with psychiatric and/or chronic brain illness have a 3–5-fold increased risk of delayed emergence and up to 60% incidence of postoperative delirium. Pathophysiological mechanisms involve GABAergic overinhibition, impaired perfusion, mitochondrial energy failure, and inflammatory amplification. Regional/local and spinal anesthesia may offer physiological advantages, preserve cerebral perfusion, and lower neurological complication rates. Conclusions: General anesthesia may exacerbate pre-existing brain vulnerability, converting reversible neural suppression into irreversible dysfunction. Therefore, whenever possible, regional/local or spinal anesthesia with or without sedation should be prioritized in those neurologically vulnerable patients to reduce the length of hospital stay (LOS) and to lower postoperative neurological complications and risks in psychiatric and neurologically unstable patients.

## 1. Introduction and Clinical Significance

Anxiety or a sense of fear is one of the most common emotional responses during the preoperative period and may intensify the physiological consequences of trauma and surgical stress. Factors such as tissue injury, blood loss, hormonal imbalance, and exposure to anesthetic agents interact with psychological stress to increase perioperative risk. In patients with psychiatric disorders or pre-existing neurological vulnerability, the combined effects of general anesthesia, altered oxygenation, and perioperative fluid shifts can further impair cerebral function and elevate the risk of postoperative neurological complications [1].

Orthopedic and hip-fracture surgeries are among the most frequent emergency procedures worldwide and represent a leading source of perioperative neurological morbidity [2]. Although anesthesia-related mortality is rare in otherwise healthy individuals, patients with psychiatric, post-stroke, or neurodegenerative disorders experience a two- to threefold increase in in-hospital mortality following orthopedic or hip-fracture surgery [2,3,4]. These individuals may also have a higher risk of postoperative delirium, coma, or prolonged unresponsiveness due to altered neural mechanisms of arousal and recovery [2,5,6]. Established risk factors for postoperative delirium include advanced age, male sex, diabetes, and preoperative cognitive dysfunction [2].

Psychiatric patients often receive long-term psychotropic therapy, which profoundly alters central and peripheral neurotransmission and modifies physiological responses to anesthetic agents [7]. Growing evidence suggests that GA amplifies both neural and systemic vulnerability in this population [8,9], whereas some observational studies suggest potential advantages of regional or spinal anesthesia in reducing stress on the vulnerable brain [10,11,12,13], but large randomized trials such as REGAIN [14] and meta-analyses [15] do not show clear outcome differences between regional and general anesthesia.

In Figure 1 below, there is a comparative demonstration of postoperative coma and death rates between general and regional anesthesia in both vulnerable and general patient groups.

The management of patients with psychiatric or neurological instability during surgery represents a great challenge to balance between surgical necessity and cerebral vulnerability [8,11]. Although advances in anesthetic techniques have markedly improved perioperative safety [11,16], they have not yet overcome the challenges of the fragile brain [8,17]. Individuals with psychiatric illness or cerebrovascular injury often present chronic disturbances in neurotransmission, perfusion, redox balance, and neuroinflammatory activity, which are the same physiological systems influenced by anesthetic agents [18,19]. When such brains are subjected to global pharmacologic suppression, the physiological consequences can be profound and occasionally irreversible [5,6].

Psychiatric disorders such as schizophrenia, bipolar disorder, and major depressive illness involve intrinsic dysregulation of GABAergic and glutamatergic neurotransmission [12,13], the same pathways targeted by common anesthetics such as propofol, sevoflurane, and isoflurane [6,17]. Chronic exposure to psychotropic medications further alters receptor sensitivity, mitochondrial function, and cerebral metabolic activity, reducing neural tolerance to additional pharmacologic suppression [8,12]. Similarly, post-stroke patients exhibit impaired autoregulation and reduced perfusion reserve [9], making even transient reductions in cerebral blood flow or oxygenation under GA potentially injurious [6,9].

Orthopedic surgery, mainly hip-fracture trauma management, provides a particular clinical relevance [14,15]. Patients frequently present emergently, are elderly, and may already be cognitively or psychiatrically compromised [3,4]. In such situations, GA is often selected for practical convenience despite the availability of regional/LA or spinal alternatives with or without sedation that preserve consciousness and maintain autoregulation [14,15,20]. The intersection of cerebral vulnerability, systemic stress, and deep sedation may therefore transform a routine procedure into a high-risk event for irreversible neurological injury [5,6].

The impact of perioperative general anesthesia on the vulnerable brain remains insufficiently understood. This review addresses this gap by summarizing current evidence on anesthesia-related neurological risks in this category of cerebrally vulnerable patients.

Thus, the objective of this review is to examine the neurological risks associated with general anesthesia in patients with psychiatric disorders, post-stroke conditions, or chronic cerebral neurodegenerative diseases undergoing orthopedic and hip-fracture surgery. By integrating clinical observations with mechanistic and epidemiological evidence, this review seeks to clarify how general anesthesia may exacerbate cerebral vulnerability in these populations, leading to delayed emergence, postoperative delirium, coma, or death. The main goal is to promote safer anesthetic strategies through interdisciplinary collaboration among anesthesiologists, psychiatrists, neurologists, and surgeons, and to encourage institutional policies that prioritize regional or spinal anesthesia whenever feasible to protect neurologically unstable patients from irreversible brain injury.

## 2. Methodology

This narrative mechanistic review integrates the author’s clinical observations with current evidence on the impact of general anesthesia on cerebrally vulnerable patients undergoing orthopedic surgery. A structured search was conducted in PubMed, Google Scholar, Scopus, and Web of Science for studies published between January 1990 and October 2025, using keywords related to anesthesia, psychiatric, neurodegenerative disorders and cerebrovascular disease, orthopedic surgery, and postoperative delirium. Clinical and experimental studies reporting outcomes such as coma, delirium, mortality, or cognitive decline were included, while studies lacking clear anesthetic or neurological data were excluded. Evidence was synthesized across five mechanistic domains: neurotransmitter dysregulation, cerebrovascular impairment, mitochondrial dysfunction, neuroinflammatory activation, and oxidative/redox imbalance. These five domains were selected because they represent the key mechanisms consistently shown to mediate anesthesia-related effects on vulnerable brain function. A representative clinical case and the author’s hospital-based observations were incorporated to illustrate ethical and physiological implications. This review used only published studies and the author’s own clinical and hospital experience, without any patient-identifying information, so ethics approval was not needed.

## 3. Clinical Perspective: A Fragile Mind in a Surgical Emergency


*The following vignette is a composite clinical scenario constructed from the author’s hospital-based observations; it does not describe any identifiable patient, and therefore, no ethics approval or consent was required.*


In an urban emergency department, a middle-aged man with chronic psychiatric illness was admitted after an accident of falling from a height and sustaining a per-trochanteric femoral fracture. He was agitated and fearful; his speech was fragmented and incoherent, but one thing he said was clear: *“No operation… I want to go home.”* No family member accompanied him. Alone, anxious, and misunderstood, he was judged incapable of informed consent. To avoid complications and to speed the surgery, the consulting psychiatrist prescribed sedatives, after which the anesthesia team proceeded with general anesthesia, fearing patient movement under spinal anesthesia. Within hours of surgery, the patient failed to regain consciousness. He entered a deep coma and died shortly afterward in the intensive care unit.

This case illustrates the intersection of biological vulnerability and system-level failure. The decision to administer GA—rather than attempting spinal or regional anesthesia or considering conservative management—overlooked both the physiological fragility of the psychiatric brain and the ethical obligation to minimize harm. What appeared to be agitation and refusal may have represented a clear, but chaotic, expression of fear and physiological exhaustion. When a dysregulated brain is subjected to global pharmacologic suppression, the final elements of homeostatic control can be lost, and anesthesia-induced coma may progress to irreversible injury.


**Research Question**



*Why should general anesthesia be avoided in cerebrally vulnerable patients?*


## 4. Pathophysiology

Pre-existing patient vulnerabilities, when compounded by critical perioperative events, can result in the development of overt neurocognitive deficits. The transition from anesthesia-induced unconsciousness to irreversible neurological failure in vulnerable patients results from the convergence of several interdependent mechanisms rather than a single insult [5,6,21]. In psychiatrically or neurologically compromised brains, general anesthetic exposure amplifies pre-existing biochemical and vascular fragility, disrupting neuronal stability across multiple levels [7,8,18]. These mechanisms are summarized in the following model (Figure 2), adapted from Tawfik et al. [21].

Chronic exposure to antipsychotics, antidepressants, or mood stabilizers alters receptor sensitivity, catecholamine reserves, and autonomic tone, predisposing these patients to exaggerated cardiovascular and neurological reactions during anesthesia [7].

At the synaptic level, over the last 20 years, neuropharmacologists have revealed that one of the most important target sites for general anesthetics is the GABA_A_ receptor [22]. Anesthetic agents potentiate GABAergic inhibition and suppress glutamatergic excitation through GABA_A_ and NMDA receptor modulation [13,17,22]. In healthy brains, this produces reversible suppression of brain activity; in diseased brains, whether under treatment of psychotropic medications or not, both the receptor imbalance and impaired glutamate handling lead to excessive inhibition, reduced excitatory drive, and delayed recovery of arousal [12,13]. Prolonged GABAergic dominance may therefore contribute to postoperative coma or delayed emergence [6,13,22].

Cerebral perfusion adds a second dimension of risk [9]. Psychiatric and post-stroke patients frequently exhibit defective autoregulation and diminished vascular responsiveness [8,9]. Volatile anesthetics further reduce cerebral blood flow by decreasing metabolic rate and systemic pressure [6,9]. Under such marginal conditions, minor hypotension or hypoxia can precipitate ischemia, particularly when neuronal metabolic capacity is already impaired, resulting in prolonged unresponsiveness [5,6,8].

At the cellular level, the majority of commonly used anesthetic agents have neurotoxic effects [23,24], inducing widespread neuronal degeneration, interfering with mitochondrial respiration and oxidative phosphorylation, mainly through inhibition of complex I of the electron transport chain [8,13,22]. The process also appears to involve activation of the oxidative stress-associated mitochondrial apoptosis pathway [23,24]. Vulnerable neurons are already operating with limited ATP due to prior injury or chronic medication exposure, and undergo further energy failure due to this additional energetic deficit [9,12]. The resulting energy crisis disrupts ion-pump function, destabilizes membrane potential, and activates apoptotic and necrotic pathways [18,19,23].

In parallel, general anesthesia activates inflammatory and oxidative pathways that further compromise neural integrity [18,19]. Elevated pro-inflammatory cytokines (IL-6, TNF-α, and IL-1β) and microglial activation disrupt the blood–brain barrier, allowing peripheral mediators to sustain neuroinflammation beyond the anesthetic period [18,19,25]. In brains already primed by psychiatric disease or vascular injury, this response becomes excessive, producing prolonged encephalopathy rather than transient sedation [8,12,24].

Cumulative redox imbalance represents a unifying mechanism linking these pathways [13,19]. Anesthetic agents increase reactive oxygen species and deplete antioxidants such as glutathione [18,19,25]. In unstable brains, this oxidative shift accelerates lipid and protein oxidation and impairs neuronal recovery [13,18,24]. Regional anesthesia may confer relative antioxidant and perfusion advantages by reducing catecholamine release, improving oxygen delivery, and limiting ischemia–reperfusion injury [20,26].

Together, these processes form a continuum of vulnerability: synaptic inhibition silences cortical activity [5,13]; perfusion failure restricts substrate delivery [9]; mitochondrial suppression exhausts energy reserves [8,13]; and inflammation with oxidative stress impedes recovery [18,19]. The outcome is a progression from reversible anesthetic suppression to irreversible neurological dysfunction [5,6]. Recognizing these mechanisms underscores the rationale for preferring regional or spinal anesthesia, which better preserves consciousness, perfusion, and redox stability in fragile brains [14,15].

## 5. Clinical Evidence and Discussion

Clinical evidence increasingly confirms that patients with psychiatric disorders, aging, or cerebrally vulnerable brains experience disproportionate adverse outcomes under general anesthesia plus the impact of surgical trauma [3,8,24,25]. Numerous cohort and retrospective studies have shown higher incidences of postoperative coma, delirium, and cognitive decline among individuals with psychiatric disorders, cerebrovascular disease, or pre-existing cognitive impairment [11,24,27]. These patients also exhibit impaired stress adaptation and dysregulated neuroendocrine and immune responses, contributing to increased perioperative morbidity and mortality [7]. Adverse intraoperative reactions such as hypotension, arrhythmia, delayed emergence, or postoperative confusion occur more frequently in this population.

In orthopedic surgery—particularly hip and femoral fracture repair—these risks are intensified by emergency presentation, advanced age, polypharmacy, and underlying neuropsychiatric comorbidities [3,4]. Some observational studies suggest possible differences in outcomes between general and regional anesthesia in high-risk patients; however, large randomized trials such as REGAIN [14] and recent meta-analyses [15] do not show clear superiority of one technique over the other. Outcomes in vulnerable patients may reflect their underlying cerebral fragility rather than the anesthetic method alone [1,10,11,20]. Although regional anesthesia has some physiological advantages, there is no consistent evidence that these benefits—such as changes in oxidative stress—translate into better outcomes for orthopedic or neurologically vulnerable patients [20]. These findings, often attributed to physical weakness or systemic illness, likely reflect underlying neural mechanisms such as impaired autoregulation, mitochondrial dysfunction, and neuroinflammatory amplification [18,19].

Case literature provides additional support [8]. Reports describe psychiatric patients—commonly treated with antipsychotics, antidepressants, or benzodiazepines—who developed prolonged unresponsiveness or fatal coma after otherwise uneventful anesthetic induction [5,6]. In many instances, no structural lesion or metabolic abnormality was identified, suggesting functional decompensation rather than anatomical injury [6,13]. Similar outcomes have been reported in post-stroke patients who exhibit delayed recovery and postoperative encephalopathy after deep anesthesia [3,9]. These observations align with experimental evidence showing that anesthetic-induced cortical suppression and mitochondrial inhibition can convert reversible pharmacologic states into irreversible neuronal dysfunction when baseline homeostasis is compromised [13,18].

Psychotropic–anesthetic interactions further compound this risk [8,12]. Chronic use of dopamine antagonists, lithium, or benzodiazepines alters receptor density, neurotransmitter turnover, and cerebral perfusion [12,13]. When anesthetic agents act on the same receptor systems, paradoxical excitation, hemodynamic instability, or excessive suppression may occur [13,17]. Abrupt discontinuation of psychotropic medications during the perioperative period can exacerbate neurochemical imbalance and increase anesthetic toxicity [8,12]. Effective preoperative coordination between psychiatry and anesthesiology is therefore essential, yet often lacking in emergency orthopedic settings [3,4].

The clinical vignette presented in this review illustrates these principles. The psychiatric patient who expressed fear and resistance prior to hip-fracture surgery underwent general anesthesia, failed to awaken postoperatively, and died shortly thereafter. This case underscores the danger of prioritizing procedural convenience over neurological safety, particularly in patients without advocates or psychiatric stabilization [3].

Beyond individual cases, these patterns highlight systemic deficiencies in perioperative decision-making [8,16]. Anesthetic selection is frequently guided by expediency rather than individualized neuropsychiatric risk assessment. Regional and spinal anesthesia represent alternatives that preserve autoregulation and consciousness, minimize neurochemical disturbance, and are associated with lower delirium rates, faster recovery, and greater hemodynamic stability [14,20,27]. For psychiatric or cerebrally unstable patients, awake or lightly sedated regional techniques provide the added advantage of continuous neurological monitoring and communication, reducing the risk of unrecognized intraoperative compromise [3,15,20]. Avoiding excessively deep levels of anesthesia in general may help reduce postoperative neurological complications. Evidence shows that maintaining an appropriate depth of anesthesia, rather than deep suppression, can lower the risk of postoperative delirium [28]. Furthermore, avoiding GA in neurologically vulnerable patients may also mitigate long-term cognitive deterioration, as deep anesthetic exposure has been linked to sustained neuroinflammatory and mitochondrial changes [18,19,20].

Together, these findings emphasize that anesthetic choice in neuropsychiatric and cerebrally vulnerable patients must extend beyond procedural practicality to encompass cerebral safety and ethical responsibility.

## 6. Clinical Recommendations and Future Directions

The synthesis of mechanistic and clinical evidence suggests that patients with psychiatric, cerebrovascular, or neurodegenerative vulnerability require a different approach to anesthesia, especially in trauma and orthopedic surgery [3,8]. In these individuals, regional or spinal anesthesia can sometimes offer a gentler option because it limits exposure to deep general anesthesia and may help maintain physiological stability. However, current clinical studies do not show clear neurological superiority of regional techniques over general anesthesia in these settings [10,11,12,13]. For this reason, the choice of anesthesia should focus on protecting the vulnerable brain through individualized risk assessment and close collaboration between psychiatry, anesthesiology, and surgery [8,16,29].

Early identification of patients at neurological risk is essential [3,8]. Anxiety may reflect underlying psychiatric pathology and influence both presentation and perioperative outcomes [30]. Individuals with severe psychiatric illness, previous stroke, cognitive impairment, or long-term psychotropic therapy should undergo targeted preoperative evaluation, including cognitive assessment, medication review, and psychiatric or neurological consultation [8,12]. Psychiatric involvement should continue throughout the perioperative course, and clinical pharmacologists should assist in managing complex drug interactions [30].

Whenever feasible, regional or spinal anesthesia can be a useful option [14,15]. These techniques preserve consciousness and cerebral autoregulation, allowing continuous neurological assessment [9]. Agitation in psychiatric patients—often prompting the use of GA—should instead be addressed through communication, reassurance, and, if needed, mild short-acting sedation [8,12]. Understanding whether agitation stems from fear, pain, or confusion is crucial [4,8]. Institutional guidelines should mandate collaborative decision-making and prioritize regional anesthesia for high-risk patients [3,16].

When GA is unavoidable, anesthesiologists must balance psychiatric stability with anesthetic safety. Continuation of psychotropic medication, avoidance of abrupt withdrawal, and prevention of pharmacologic interactions are essential to reduce neurochemical destabilization [7]. Neuroprotective measures should include minimal effective dosing, EEG-based depth monitoring [28], maintenance of cerebral perfusion and oxygenation, and preference for agents with favorable cerebral profiles such as dexmedetomidine or low-dose ketamine [6,9]. Postoperative care must include vigilant neurological monitoring and prompt management of delayed emergence or delirium [11,27].

Ethical considerations are integral to perioperative decision-making [8,16]. Agitation or refusal should not be dismissed as non-compliance but understood as potential expressions of distress or residual awareness [8,12]. In the absence of patient advocates, decisions to proceed with GA require multidisciplinary justification [3,4]. Hospitals should implement standardized protocols addressing cerebral vulnerability, including preoperative checklists that flag neuropsychiatric risk factors, mandate consideration of regional anesthesia, and document the rationale when GA is chosen [14,15]. Training programs should incorporate education on the neurobiology of psychiatric illness and ethical management of cognitively unstable patients [8,12].

Future research must progress from observational correlations to mechanistic validation [18,19]. Standardized redox and oxidative biomarker panels, along with multimodal neuromonitoring techniques such as near-infrared spectroscopy, processed EEG, and microcirculatory imaging, are needed to elucidate how anesthetic techniques influence neuronal redox balance and to define thresholds for neurotoxicity [6,17,26]. Experimental studies focusing on mitochondrial resilience, neuroinflammation, and redox homeostasis may identify novel neuroprotective strategies [18,19]. Qualitative research into communication and consent in vulnerable patients remains important [8,16]. Additionally, integrated risk-prediction models incorporating modifiable and non-modifiable factors should be developed to improve perioperative outcomes in orthopedic surgery involving neurologically fragile patients [2].

Ultimately, both the evidence and clinical reality converge on a single principle: when the brain is unstable, keep it awake. This approach represents a shift from procedural convenience to cerebral stewardship. Preserving consciousness in the fragile brain is not passive avoidance but active protection, safeguarding neural integrity and human dignity where deep anesthesia too often results in irreversible loss [5,6].

## 7. Conclusions

Caring for patients with psychiatric or neurologically fragile brains during surgery is different from caring for healthy patients. General anesthesia is safe for most people, but these vulnerable patients do not react to it in the same way. Their brain chemistry, blood flow, and stress responses are already unstable. When exposed to deep general anesthesia, they are at higher risk of delayed awakening, prolonged confusion, or even serious neurological decline.

This does not mean that general anesthesia is unsafe for everyone. It means that in patients with psychiatric illness, cognitive disorders, or previous brain injury, the margin of safety is much smaller. These patients require special attention, careful planning, and anesthesia that protects their already fragile brain function.

Whenever possible, regional, local, or spinal anesthesia should be considered first. These techniques help maintain consciousness, support cerebral blood flow, and avoid exposing the vulnerable brain to deep suppression. When general anesthesia cannot be avoided, it should be given with close monitoring of anesthetic depth, stable blood pressure, and careful management of psychiatric medications.

The case described in this review reminds us of what can happen when these vulnerabilities are overlooked. It reflects real situations that clinicians see in daily practice, where a combination of psychiatric instability, emergency surgery, and deep anesthesia leads to tragic outcomes.

The message is simple and important: vulnerable brains need protection. Our anesthetic choices must respect this fragility to prevent avoidable harm.

This recognition demands a redefinition of anesthetic reasoning and ethics. The routine use of general anesthesia for orthopedic emergencies should be replaced with approaches tailored to each patient, taking into consideration how vulnerable their brain is and how that affects the outcome. Regional/local and spinal anesthesia with or without sedation should be preferred to shorten hospital stay, decrease the risk of postoperative complications, and preserve consciousness and cerebral perfusion in such vulnerable patients. When GA is unavoidable, it must follow strict neuroprotective principles with vigilant monitoring.

The tragedy of the psychiatric patient who entered surgery against his expressed distress and never regained consciousness remains a reminder that the unstable brain demands caution, empathy, and restraint. The lesson is both scientific and moral: when the brain is unstable, keep it awake.

## Figures and Tables

**Figure 1 reports-08-00263-f001:**
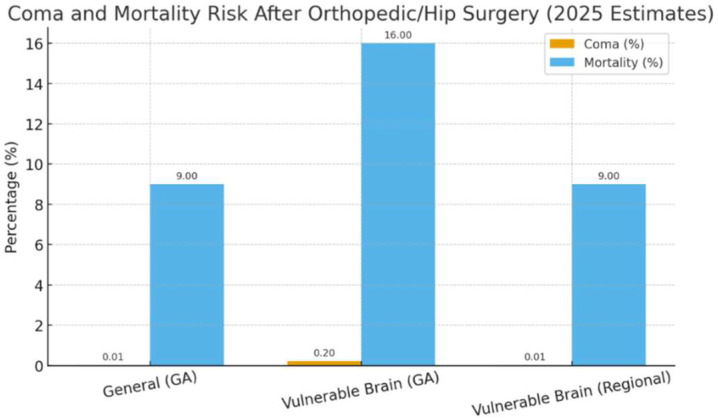
Estimated comparative risks of postoperative coma and in-hospital mortality after orthopedic and hip-fracture surgery. *This figure is a conceptual illustration intended only to demonstrate relative patterns reported in the literature. The numerical values are schematic, illustrative conceptual values, not pooled estimates. They are derived qualitatively from the general ranges described in references* [3,4,10]. The bar chart illustrates the incidence of postoperative coma and mortality in the general surgical population and in psychiatric or neurologically vulnerable patients under general and regional anesthesia. Data were synthesized from published epidemiological studies [3,4,10] and international anesthesia safety reports. Figure created by the author.

**Figure 2 reports-08-00263-f002:**
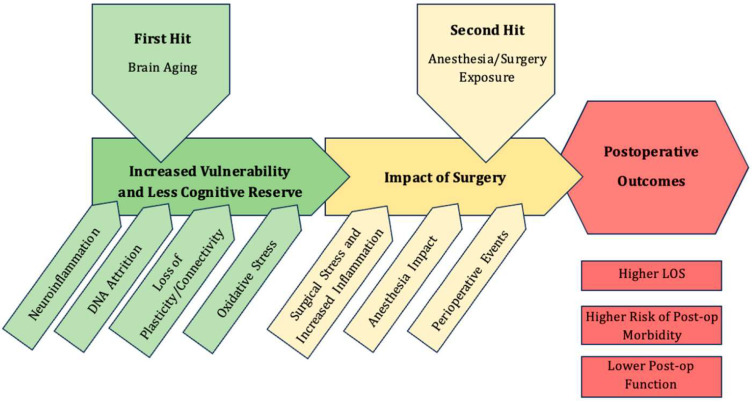
Two-hit model of perioperative neurocognitive disorders (PNDs), highlighting the pathophysiology of brain aging or cerebral vulnerability, which reduces the cognitive reserve, the critical perioperative events, and the postoperative trajectory. Adapted from Tawfik et al. [21].

## Data Availability

The original contributions presented in this study are included in the article. Further inquiries can be directed to the corresponding author.

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
