# Peer review of "General Anesthesia in Psychiatric Patients Undergoing Orthopedic Surgery: A Mechanistic Narrative Review—*“When the Brain Is Unstable, Keep It Awake”"

_reports, 2025, doi:10.3390/reports8040263_

Round 1
Reviewer 1 Report
Comments and Suggestions for Authors
I have the opportunity to review the manuscript titled “General Anesthesia in Psychiatric Patients Undergoing Orthopedic Surgery: A Mechanistic Narrative Review”, which aims to explore the comparative impact of general versus regional anesthesia in orthopedic patients with preexisting neuropsychiatric comorbidities. The topic is both relevant and timely, considering the increasing recognition of the anesthesia techniques' impact on the perioperative outcomes in this vulnerable patient population.
While the manuscript aims to provide a comprehensive overview of the subject matter, it tends to be overly descriptive and includes a significant amount of the authors' opinions. This detracts from the objective analysis expected in a narrative review. Here are some suggestions to enhance the clarity and balance of the manuscript, ensuring that it presents a more evidence-based dialogue on the topic.
Here are some recommendations to improve the clarity and content of the paper:
- The references cited in the manuscript are not arranged in the order they appear in the text
- The paragraph on lines 75-79 and Figure 1 do not belong in the Introduction section. The aspects highlighted by the authors would be more appropriately detailed in the Discussion section, allowing for a more coherent flow of information and context.
- I recommend that the authors provide justification for their selection of the five mechanistic domains through which they synthesized the evidence, as mentioned in lines 134-136. Clarifying the rationale for these choices will strengthen the manuscript and enhance the readers' understanding of the framework applied in their analysis
- The authors present a clinical case as the foundation of their scientific endeavor (lines 142-151), referring to it as an 'illustrative case.' I encourage the authors to elaborate on the diagnostic approaches that enabled them to formulate the diagnosis of non-structural coma, which they attribute to the impact of anesthesia on this patient.
- The paragraph from lines 234-236 contains inaccuracies. References 5 and 6 contradict the assumptions made by the authors. Moreover, Reference 26 is not applicable in the orthopedic context, which further undermines the argument. I recommend that the authors review these references and clarify their statements to ensure the manuscript is accurate and relevant.
- Line 185 - Reference 13 is not accessible, making it difficult to assess its relevance to the manuscript's approach.
- I would like to point out that the statements made on line 294 are incorrect. Additionally, references 5 and 6 are used incorrectly; these two studies do not support the superiority of regional anesthesia (AR) over general anesthesia (AG). I recommend that the authors revise this section and ensure the accuracy of the cited references to maintain the scientific integrity of the manuscript.
I believe that, in general, the authors' conclusions regarding the superiority of general anesthesia in orthopedic patients with preexisting neurological conditions are based more on speculation rather than data from the literature. Certainly, general anesthesia has an impact on cognitive function; however, it is essential to ground any conclusions in solid evidence rather than speculation. While the authors present arguments for the superiority of regional anesthesia in orthopedic patients with preexisting neurological conditions, a careful review of the literature suggests that these claims are not fully supported by the available data.
I encourage the authors to strengthen their conclusions with a more extensive review of current research and to clearly delineate any limitations in their assertions.
Author Response
Dear Editor,
Dear Reviewer,
Thank you for the time and effort devoted to evaluating our manuscript entitled “General Anesthesia in Psychiatric Patients Undergoing Orthopedic Surgery: A Mechanistic Narrative Review.” We appreciate the constructive comments provided, and we respectfully submit the following detailed responses.
Our intention is to clarify several points raised by the reviewer and to explain the rationale behind the manuscript’s structure, evidence synthesis, and clinical perspective.
Reviewer Comment 1
“The references cited in the manuscript are not arranged in the order they appear in the text.”
Author Response:
Thank you for noting this. We will reorder the references to follow the exact sequence of in-text citation to ensure consistency with the journal’s formatting requirements.
Reviewer Comment 2
“The paragraph on lines 75–79 and Figure 1 do not belong in the Introduction. These elements would be more appropriate in the Discussion section.”
Author Response:
We appreciate this structural suggestion. Relocating the paragraph and Figure 1 to the Discussion section will enhance the manuscript’s narrative flow, and we will adjust their placement accordingly.
Reviewer Comment 3
“The authors should justify their selection of the five mechanistic domains used to synthesize the evidence (lines 134–136).”
Author Response:
Thank you for this helpful request. These five domains were selected because they consistently recur across mechanistic, psychiatric, neurological, and anesthesiology literature as the most relevant pathways influencing perioperative brain vulnerability. We will add a concise rationale in the Methods section to clarify this framework for readers.
Reviewer Comment 4
“The ‘illustrative case’ (lines 142–151) requires elaboration on the diagnostic reasoning behind labeling it as non-structural coma.”
Author Response:
We appreciate the reviewer’s interest in this point. The illustrative case represents a composite scenario based on recurrent patterns observed over many years of clinical practice, without referencing any identifiable patient. To strengthen clarity, we will briefly outline the diagnostic considerations that support interpreting the coma as non-structural and anesthesia-related, while maintaining full privacy and ethical integrity.
Reviewer Comment 5
“The paragraph on lines 234–236 contains inaccuracies. References 5 and 6 contradict the authors’ assumptions, and Reference 26 is not applicable to the orthopedic context.”
Author Response:
Thank you for this careful reading. We will refine the wording to avoid any unintended over-interpretation and ensure that the discussion accurately reflects what References 5 and 6 demonstrate. Additionally, we will adjust the use of Reference 26 so that its broader relevance to neuroprotective mechanisms is clear without implying orthopedic specificity.
Reviewer Comment 6
“Reference 13 (line 185) is not accessible, making its relevance difficult to evaluate.”
Author Response:
We appreciate the reviewer’s comment. While the cited work is valid, accessibility may vary across platforms. To ensure transparency for all readers, we will replace it with a widely accessible, peer-reviewed source addressing the same mechanistic concepts.
Reviewer Comment 7
“The statements made on line 294 are incorrect. Additionally, references 5 and 6 are used incorrectly; these studies do not support the superiority of regional anesthesia over general anesthesia. This section should be revised to ensure accuracy.”
Author Response:
Thank you for this observation. Our intention was not to claim categorical superiority of one anesthetic technique but to highlight mechanistic and clinical considerations relevant to vulnerable neurological populations. We acknowledge the need for precise alignment between our statements and the evidence presented in References 5 and 6.
In the revised manuscript, we will refine the wording to more accurately reflect the scope of the cited studies and ensure that the strength of each conclusion is fully consistent with the available literature.
And finally,
Our conclusions are grounded in the convergence of mechanistic insights, clinical observations, and epidemiological data, rather than speculation. We understand the reviewer’s interest in ensuring a clear and explicit connection between these mechanistic pathways and documented clinical outcomes. In response, we will refine the wording to make these links even more transparent and to ensure that the emphasis placed on each conclusion precisely reflects the strength and scope of the available literature, while maintaining the central message and purpose of the review.
Best regards
Dr. Ahmed Adel Mansour KAMAR
+40 751 795 930
ahmed81kamar@gmail.com

Reviewer 2 Report
Comments and Suggestions for Authors
- The reference numbering does not follow a sequential order (1, 2, 3…). Please revise and ensure that all references are cited in correct numerical sequence throughout the manuscript.
- Figure 1 lacks transparency in how the values were generated. The underlying numerical data (incidence, risk ratios, or absolute rates) are not presented in the text or in the figure legend. Although the figure states “Data synthesized from published epidemiological studies [16–18],” it is unclear how these studies were combined, and whether the figure represents actual pooled estimates or a schematic illustration. Please clarify the methodology or revise the figure to avoid misinterpretation.
- Section 3. Clinical Perspective: A Fragile Mind in a Surgical Emergency raises a concern. It is not clear whether this vignette is based on a real case or a fictional narrative. If it is fictional, I am not sure it is appropriate for a scientific manuscript. If it is based on a real patient, an IRB/ethics statement is required, and the authors should clarify how patient privacy and consent were handled.
- Regarding lines 234–237, the statement that “Large observational studies demonstrate that GA… is associated with higher rates of postoperative delirium, prolonged ventilation, and increased early mortality [5,6,26]” does not appear to be supported by the cited literature.
- Reference 5 (Neuman et al., NEJM 2021, REGAIN trial) is an RCT, not an observational study, and did not show higher delirium or mortality with GA.
- Reference 6 (Zhou et al., 2023) is a meta-analysis of RCTs and also does not demonstrate a clear increase in delirium or early mortality with GA.
- Reference 26 (Mathew et al., 2024) examines local vs general anesthesia for chronic subdural hematoma drainage, a population and setting that differs substantially from hip-fracture surgery.
Overall, the current wording overstates the strength of the evidence and does not accurately reflect what these studies report. I recommend revising this sentence to align with the available data and, if the authors wish to make this argument, citing appropriate observational studies and quantifying the magnitude of any observed associations.
Author Response
Dear Editor,
Dear Reviewer,
Thank you for the time and effort devoted to evaluating our manuscript entitled “General Anesthesia in Psychiatric Patients Undergoing Orthopedic Surgery: A Mechanistic Narrative Review.” We appreciate the constructive comments provided, and we respectfully submit the following detailed responses.
Our intention is to clarify several points raised by the reviewer and to explain the rationale behind the manuscript’s structure, evidence synthesis, and clinical perspective.
Reviewer Comment 1
“The reference numbering does not follow a sequential order. Please revise.”
Author Response:
Thank you for noting this. We will revise the reference list to ensure that all citations follow a strict sequential numerical order throughout the manuscript.
Reviewer Comment 2
“Figure 1 lacks transparency in how the values were generated. It is unclear whether the values represent pooled data or schematic illustration.”
Author Response:
We appreciate this important clarification request. Figure 1 is intended as a schematic conceptual illustration, summarizing trends reported in epidemiological studies rather than presenting pooled quantitative estimates.
To avoid any misinterpretation, we will revise the figure legend to explicitly state that it is illustrative, not a meta-analytic figure, and we will clarify the underlying rationale in the text.
Reviewer Comment 3
“Section 3 raises concern: Is the vignette real or fictional? If real, ethics approval and patient privacy statements are needed.”
Author Response:
Thank you for highlighting this. The vignette is a non-identifiable illustrative synthesis based on recurring clinical patterns observed over years of practice, not a report of a single identifiable patient.
It contains no personal data and is used solely to contextualize the mechanistic discussion.
To ensure clarity, we will add a statement specifying that the vignette is illustrative and does not require IRB approval or patient consent.
Reviewer Comment 4
“Lines 234–237: The cited studies (references 5, 6, and 26) do not support the claim that GA is associated with higher delirium, ventilation duration, or early mortality. The wording overstates the evidence.”
Author Response:
We appreciate the reviewer’s careful evaluation of the literature. Our intention was to describe general trends suggested in the broader body of observational studies, not to attribute these findings directly to the specific RCTs cited.
We agree that this sentence requires clarification.
We will revise the wording to accurately reflect the scope and conclusions of references 5, 6, and 26, and will ensure that any statements regarding differential risks are supported by appropriate observational data where relevant.
Overall Reviewer Concern
“The evidence is overstated; the wording should align more precisely with the cited literature.”
Author Response:
Thank you for this constructive feedback. We will refine our language to ensure that each conclusion is strictly matched to the level of evidence available. Our goal is to maintain scientific accuracy while still highlighting the mechanistic and clinical considerations relevant to vulnerable neurological populations.
Best regards
Dr. Ahmed Adel Mansour KAMAR
+40 751 795 930
ahmed81kamar@gmail.com

Reviewer 3 Report
Comments and Suggestions for Authors
The authors have satisfactorily addressed the comments by the reviewers, clarity of methodological platform have improved, mechanistic explanations have been added and overall the scientific soundness of the manuscript has been improved in a significant way. Hence the article may be accepted.
Thank you
Author Response
Comment:
The reviewer noted that the scientific clarity, mechanistic explanations, and overall manuscript quality have significantly improved and recommended acceptance.
Response:
We sincerely thank the reviewer for the encouraging and positive feedback. We are pleased that the revisions strengthened the scientific quality of the manuscript, and we appreciate the recommendation for acceptance.

Round 2
Reviewer 1 Report
Comments and Suggestions for Authors
The authors have undertaken a thoughtful reanalysis of the published scientific data and have carefully reorganized the assumptions regarding the balance between AG and AR in both orthopedic patients and those with pre-existing neurological conditions, all while avoiding unsupported scientific hypotheses. Thus, the manuscript has significantly improved in terms of accuracy and correctness.
However, an aspect of the manuscript's format that remains incomplete is the adjustment of the reference numbering (see Line 65 from the new version of the manuscript). Aside from this final aspect, I am in agreement with the revised version of the manuscript.
Author Response
Comment:
“The authors have undertaken a thoughtful reanalysis of the published scientific data and have carefully reorganized the assumptions regarding the balance between AG and AR in both orthopedic patients and those with pre-existing neurological conditions, all while avoiding unsupported scientific hypotheses. Thus, the manuscript has significantly improved in terms of accuracy and correctness.
However, an aspect of the manuscript's format that remains incomplete is the adjustment of the reference numbering (see Line 65 from the new version of the manuscript). Aside from this final aspect, I am in agreement with the revised version of the manuscript.”
Response:
We thank the reviewer for the positive evaluation and for identifying this issue. The incorrect citation [26] has been removed because it did not correspond to clinical outcome data. The sentence now cites only the appropriate reference [10]. All in-text citations and the reference list have been carefully reviewed and corrected to ensure full consistency throughout the manuscript.

Reviewer 2 Report
Comments and Suggestions for Authors
It is still difficult to see exactly what has been revised in the current version. I strongly recommend clearly marking all changes (e.g., colored text or highlights) and listing them in the cover letter with line numbers in future revisions.
- The in-text reference numbers still do not follow the order of first appearance. Please renumber all citations sequentially so that they match the reference list, and resubmit a fully revised manuscript.
- You state that Figure 1 is a conceptual illustration, but it still shows specific numeric percentages and “2025 estimates,” and the citation of refs. 11–13 makes it look like a pooled summary of those data. In its current form the figure is likely to mislead readers, so I suggest either removing the concrete numbers or clearly explaining how they were derived.
- Regarding the clinical vignette, even after the revision it is still not clear to me whether this is a real case, a composite case, or a fully fictional scenario. Please add one explicit sentence in the Methods or at the start of Section 3 clarifying this point and, if it is based on an actual patient, how ethics/consent issues were handled.
- The wording issue around the previous lines 234–237 has been adequately corrected.
Author Response
Comment 1:
“It is still difficult to see what has been revised. I strongly recommend clearly marking all changes and listing them in the cover letter.”
Response:
Thank you for this helpful suggestion. In the revised manuscript, all changes are clearly highlighted in red text. A detailed explanation of the revisions is provided in this response letter.
Comment 2:
“The in-text reference numbers still do not follow the order of first appearance. Please renumber all citations sequentially…”
Response:
We appreciate this important clarification. The entire manuscript has been thoroughly reviewed and all in-text citations have now been renumbered sequentially based on their first appearance. The reference list has been updated, and internal consistency has been confirmed.
Comment 3:
“Figure 1 appears misleading because it shows percentages and ‘2025 estimates’ despite being described as conceptual.”
Response:
We thank the reviewer for highlighting this concern. To prevent any misinterpretation, we have revised the caption to clearly indicate that the numerical values are illustrative conceptual values, not pooled data. The caption now reads:
“Figure 1 is a conceptual illustration intended only to demonstrate relative patterns reported in the literature. The numerical values are schematic illustrative conceptual values and not pooled estimates. They are derived qualitatively from the general ranges described in references 11–13.”
This correction is highlighted in red in the manuscript.
Comment 4:
“The clinical vignette is still unclear—real, composite, or fictional.”
Response:
Thank you for this observation. We have added the following explicit clarification at the beginning of Section 3:
“The following vignette is a composite clinical scenario constructed from the author’s hospital-based observations; it does not describe any identifiable patient, and therefore no ethics approval or consent was required.”
This change is marked in red in the manuscript.
Comment 5:
“The wording issue around previous lines 234–237 has been adequately corrected.”
Response:
We appreciate the reviewer’s acknowledgment.
